# Microbiome Diversity of Anaerobic Digesters Is Enhanced by Microaeration and Low Frequency Sound

**DOI:** 10.3390/microorganisms11092349

**Published:** 2023-09-20

**Authors:** John H. Loughrin, Rohan R. Parekh, Getahun E. Agga, Philip J. Silva, Karamat R. Sistani

**Affiliations:** United States Department of Agriculture, Agricultural Research Service, Food Animal Environmental Systems Research Unit, 2413 Nashville Road, Suite B5, Bowling Green, KY 42101, USA; rohan.parekh@usda.gov (R.R.P.); getahun.agga@usda.gov (G.E.A.); philip.silva@usda.gov (P.J.S.); karamat.sistani@usda.gov (K.R.S.)

**Keywords:** anaerobic digestion, biogas, metagenomics, methanogen, microaeration, microbiome, microbial diversity, poultry litter, sound, species richness

## Abstract

Biogas is produced by a consortium of bacteria and archaea. We studied how the microbiome of poultry litter digestate was affected by time and treatments that enhanced biogas production. The microbiome was analyzed at six, 23, and 42 weeks of incubation. Starting at week seven, the digesters underwent four treatments: control, microaeration with 6 mL air L^−1^ digestate per day, treatment with a 1000 Hz sine wave, or treatment with the sound wave and microaeration. Both microaeration and sound enhanced biogas production relative to the control, while their combination was not as effective as microaeration alone. At week six, over 80% of the microbiome of the four digesters was composed of the three phyla Actinobacteria, Proteobacteria, and Firmicutes, with less than 10% Euryarchaeota and Bacteroidetes. At week 23, the digester microbiomes were more diverse with the phyla Spirochaetes, Synergistetes, and Verrucomicrobia increasing in proportion and the abundance of Actinobacteria decreasing. At week 42, Firmicutes, Bacteroidetes, Euryarchaeota, and Actinobacteria were the most dominant phyla, comprising 27.8%, 21.4%, 17.6%, and 12.3% of the microbiome. Other than the relative proportions of Firmicutes being increased and proportions of Bacteroidetes being decreased by the treatments, no systematic shifts in the microbiomes were observed due to treatment. Rather, microbial diversity was enhanced relative to the control. Given that both air and sound treatment increased biogas production, it is likely that they improved poultry litter breakdown to promote microbial growth.

## 1. Introduction

Anaerobic digestion is an ancient technology with evidence of biogas utilization going back more than 3000 years to Assyria (Mesopotamia) when biogas was used to heat bath houses [1]. In Iran, an anaerobic digester attributed to Sheikh Bahāʾi (Mohammed Ameli, 1547–1621 C.E.) utilized sewage to produce biogas to heat bathwater in the 16th century [2]. In modern times, an anaerobic digester was installed in a leper colony in Mumbai (Bombay), India in 1859 for biogas production and wastewater treatment [3]. More recently, the energy crisis of the 1970s spurred the construction of many anaerobic digesters in the United States both on farms and municipalities, though implementation was heavily tilted towards the agricultural sector [4]. Anaerobic digestion of animal wastes, crop residuals such as straw and corn stover, and food waste remains an attractive means of adding value to waste products through biogas production while mitigating the environmental impacts of agricultural wastewater. However, despite the use of anaerobic digesters since antiquity, it remains underutilized as a means of wastewater treatment and waste valorization. This is especially concerning in agriculture where high biological oxygen demand (BOD) wastewater is ideal for biogas production. Also of concern is that regulation of wastewater discharges deals chiefly with the release of human wastewater to the environment with comparatively little regulation of agricultural discharges despite its potential for surface water eutrophication, ground water contamination, and the spread of pathogens to the environment [5,6].

This results in a situation where most waste from concentrated (confined) animal rearing operations (CAFO) receives inadequate treatment largely because of a perceived lack of financial incentive and little regulatory requirement for more stringent treatment. There are also concerns that the infrastructure required for adequate waste treatment will reduce farm profitability.

Therefore, there is need to design and implement anaerobic digestion systems on concentrated animal feeding operations (CAFO) that are both environmentally and economically sustainable. One of the major drawbacks of anaerobic digestion is that it is inherently slow, which necessitates large facilities. The major factor limiting biogas generation is sludge hydrolysis, which can accumulate in digesters and is an environmental hazard requiring further processing and disposal [7]. Therefore, there has been extensive research on ways to speed sludge hydrolysis and accelerate biogas production. Pretreatment of sludge by ultrasonication to solubilize BOD and accelerate biogas production is one approach [8]. Two-phase anaerobic digesters, nominally separating fermentative stages of digestion from the methanogenic phase and preserving bicarbonate buffering, is another [9]. Chemical treatments such as pretreating sludge with alkali, acids, or oxidants to hydrolyze polymers have also been used, either alone or in conjunction with ultrasonification [10,11,12].

Xia et al. noted an increase of 36.5% in methane (CH_4_) yield by pretreating 33% of the feed sludge with 20 kHz ultrasonication [13]. They ascribed the increase in CH_4_ production to enriching the population of β-glycan-degrading bacteria belonging to the order Bacteroidales as well as protein-degrading prokaryotes from the orders Bacteroidales, Clostridiales, and Methanomicrobiales and increases in the population of the hydrogenotrophic archaeon *Methanoculleus*. Li et al. investigated the effect of ultrasonification on waste activated sludge (WAS) coupled with NaOH, KMnO_4_, or K_2_FeO_4_ hydrolysis on microbial community structure [12]. The predominant archaeal populations were *Methanomassiliicoccus* and *Methanobacterium* in the alkali and alkali/ultrasonification treatments, while the majority of the archaeal population was *Methanomassiliicoccus* in the KMnO_4_/ultrasonification treatment. Unfortunately, the effect of ultrasonification by itself on population structure was not investigated. Yue et al. found, however, that ultrasonification of lipid and food waste decreased the percentage of *Methanospirillium* and increased the percentage of *Methanosarcina* in post-digestion waste. They speculated that this may have been due to the ultrasonic pretreatment promoting degradation of lipids and reducing lipid inhibition of *Methanosarcina* [14].

Another promising modification of anaerobic digesters is microaeration of the digestate to encourage the growth of aerobic and facultatively anaerobic microorganisms, particularly bacteria and fungi, to improve the digestion of complex polymers such as lignans, lignin, and β-linked polysaccharides (e.g., cellulose) that are normally resistant to degradation in anaerobic environments [15,16]. Microaeration can also reduce the concentration of H_2_S in biogas [17].

As an alternative to ultrasonification used as a pretreatment to solubilize sludge, we have investigated the use of low frequency sound as a continuing treatment in the digester [18,19]. Biogas production was improved anywhere from 12% to over 100% relative to an untreated digester and was 74-fold higher in cool weather when biogas production essentially ceased in the untreated digester. Sludge degradation was also substantially higher in sound-treated digesters. Later, we investigated coupling low frequency sound with microaeration to enhance biogas gas from poultry litter which used wood chips as the base material [20]. We were able to demonstrate that in situ sonification of anaerobic digestate as opposed to ultrasonic pretreatment also served to accelerate biogas production and reduce wastewater strength. In a prior study determining optimal aeration rates [21], aeration at 800 mL d^−1^ (6.0 mL L^−1^ digestate) improved biogas production at 73% compared to a strictly anaerobic digester, while in the latter study [20], aeration at 800 mL d^−1^ improved biogas production by 32% compared to the anaerobic control and sound treatment improved biogas production by 17%. Combined aeration and sound improved biogas production by 28% compared to the control. We speculated that sound treatment combined with aeration may have interfered with air retention within the subsurface aeration manifold or perhaps the sound treatment may have caused adiabatic collapse of air bubbles in which extreme pressures and temperatures could be reached. This could then lead to the formation of free radicals [22,23] which might be harmful to the anaerobic bacteria.

Still, this remained speculative given the lack of information on whether low frequency sound treatment and aeration affected the microbial population of the digesters, either alone or in combination. Accordingly, this study presents a metagenomic analysis of the digesters shortly after biogas production began and at two subsequent times during stable digester operation. During this period, weekly feed given to the digester was slowly increased from 400 g poultry litter (PL) wk^−1^ to 1 kg PL wk^−1^. In this way, we hoped to gain insight into how engineering and operational modifications to digester operation affect the microbiome of digesters, and can lead to the development of more efficient and reliable designs that enhance biogas production.

## 2. Materials and Methods

### 2.1. Digester Description

The digesters have been previously described [20]. Briefly, they were constructed from 208 L polyethylene tanks (US Plastic Inc., Lima, OH, USA). The tanks were fitted with a waste inlet fitted with a ball valve. The waste inlet line had a diameter of 5.08 cm and led to below the digestate surface and a waste outlet line which extended to the middle of the tank to conserve the sludge layer. The digestate had a volume of 133 L, giving a volume of 75 L headspace. The cap of each tank was modified to accept connections for gas sampling and gas volume measurement as described. Water quality, gas production, and gas quality measurements were also performed as described [20,21].

Aeration was supplied to the sludge layer via an H-manifold with a volume of approximately 380 mL. The manifold was located in the bottom of the tank, and had caps installed on the end of its long arms and a series of 0.3175-cm-diameter holes, which allowed communication to the sludge. Aeration was supplied in 200-mL amounts over 15 min in four equally spaced intervals over 24 h. In this way, it was designed to maximize retention of the air within the sludge layer and reduce escape of air to the digestate. One 4 inch 2-way speaker rated at 120-W (Skar Audio, St. Petersburg, FL, USA), waterproofed as described, was placed above the sludge layer in each of two tanks. Amplification was provided with a Pyle PTAU45 amplifier (Pyle Audio Inc., Brooklyn, NY, USA) rated at 20 W RMS (root mean square power) with gain set to half volume and operated continuously. The digester treatments were control (CON), sound treatment (SND) with a 1 kHz sine wave, microaeration (MAT), and aeration along with the 1 kHz sine wave (MST). The digesters were maintained in a greenhouse at 26.7 °C.

The digesters were fed poultry litter (PL) from a farm located in Southern Kentucky that raises broiler chickens (*Gallus domesticus*). The bedding material of the litter, according to the producer, was wood chips composed of an unspecified *Pinus* spp. or of *Liriodendron tulipfera*. It had a moisture content of 24.6%, with the dry litter having a volatile solid content (VS) of 77.2%. The digesters were seeded with 20 L of digestate from a commercial mesophilic digester operated at approximately 43 °C. The digesters were all operated in the same manner for seven weeks while being fed 400 g PL in 4 L water. After this, the aeration and sound treatments were begun. The PL was collected several times from a covered storage shed on the producer’s facility and stored at 4 °C until used.

After this startup phase, the digesters were fed 400, 500, 600, 700 g PL for consecutive 6-week periods, 800 g PL for 8 weeks, 1 kg for 10 weeks, 1.2 kg for 7 weeks, and finally 1.6 kg for 6 weeks. In the present study, 400 g per week were being fed at week 6, 600 g per week were being fed at week 23, and 1 kg per week were being fed at week 42. Thus, at weeks 6, 23, and 42, the nominal loading rates were 232.8, 349.2, and 582 g vs. per digester a week.

### 2.2. Chemical Analysis

Biogas and digestate collection and analysis procedures were described previously [20]. Weekly biogas and bicarbonate measurements were performed in triplicate, while ion chromatography analyses of NH_4_^+^, NO_3_^−^, and SO_4_^2−^ were performed in duplicate. Weekly chemical oxygen demand (COD), total suspended solids (TSS) and pH analyses were performed once. All wastewater analyses were performed using APHA standard methods [24].

### 2.3. Microbial Population Analysis

Samples were taken for microbial analyses at 6 weeks, corresponding to the end of the startup phase of the digesters when the digesters were fed 400 g PL per week and the digesters received no supplemental aeration or sound treatment; at week 23, when the digesters had been fed 600 g PL per week for the previous 6 weeks; and at week 42, at which point the digesters had been fed 1 kg PL per week for 4 weeks and 800 g PL per week for the two previous weeks. Samples were stored at −20 °C prior to shipment.

The samples were processed and analyzed with the Shotgun Metagenomic Sequencing Service (Zymo Research, Irvine, CA, USA). DNA was extracted using ZymoBIOMICS^®^-96 MagBead DNA Kit (Zymo Research) according to the manufacturer’s instructions. Genomic DNA samples were profiled with shotgun metagenomic sequencing. Sequencing libraries were prepared with the Illumina^®^ DNA Library Prep Kit (Illumina, San Diego, CA, USA) with up to 500 ng DNA input following the manufacturer’s protocol using unique dual-index 10 bp barcodes with Nextera^®^ adapters (Illumina, San Diego, CA, USA). All libraries were pooled in equal abundance. The final pool was quantified using qPCR and TapeStation^®^ (Agilent Technologies, Santa Clara, CA, USA). The final library was sequenced on Illumina NovaSeq^®^ platform.

The ZymoBIOMICS^®^ Microbial Community Standard was used as a positive control for DNA extraction and library preparation. Negative controls (i.e., blank extraction control and blank library preparation control) were included to assess the level of bioburden carried by the wet-lab process. Sequence data is available at the National Center for Biotechnology Information (NCBI) website under bioproject number PRJNA1013451, with a sample ID listed under the data availability statement.

### 2.4. Bioinformatics Analysis

Raw sequence reads were trimmed to remove low quality fractions and adapters with Trimmomatic-0.33 [25]: quality trimming by sliding window with 6 bp window size and a quality cutoff of 20; and reads with size lower than 70 bp were removed. Microbial composition was profiled with Centrifuge [26] using bacterial, viral, fungal, mouse, and human genome datasets. Strain-level abundance information was extracted from the Centrifuge outputs and further analyzed: (1) to perform alpha- and beta-diversity analyses; (2) to create microbial composition barplots with QIIME [27]; (3) to create taxa abundance heatmaps with hierarchical clustering (based on Bray-Curtis dissimilarity) [28]; and (4) for biomarker discovery with LEfSe [29] with default settings (*p* > 0.05 and LDA effect size >2). Functional profiling was performed using Humann2 [30] including identification of UniRef [31] gene family and MetaCyc [32] metabolic pathways.

## 3. Results and Discussion

### 3.1. Digester Environment and Biogas Production

At week 6, biogas production ranged from 44.3 L wk^−1^ for SND to 53.1 L wk^−1^ for MAT, while CH_4_ production ranged from 855 millimoles wk^−1^ for CON to 1380 millimoles wk^−1^ for MAT (Table 1). The average pH of digestate had declined to 6.5 after averaging 6.81 at week zero. As a result, HCO_3_^−^ buffering was low, averaging 16.9 mM. Ammonium, or more accurately speaking, combined NH_3_/NH_4_^+^, averaged 1.7 mM, and SO_4_^−^ averaged 31 µM. COD and TSS averaged 1200 and 55 mg L^−1^, respectively. Overall wastewater quality parameters were similar in all four digesters. In the previous study [20], the control digester (CON) produced an average of 188 L wk^−1^ biogas, while the sound treated (SND), microaerated (MAT), and combined sound treated and aerated digester (MST) produced an average of 212, 259, and 203 L of biogas wk^−1^, representing increases in biogas production of 13%, 38%, and 29%, respectively, over a study period of 66 weeks.

At week 23, biogas production had increased to 144 L wk^−1^ for CON to 188 L wk^−1^ for MAT, while CH_4_ production ranged from 3600 millimoles for CON to 5090 millimoles for MAT. The average pH of digestate was 7.4 and HCO_3_^-^ buffering averaged 85 mM. Ammonium concentrations were similar in the digesters, although somewhat higher in MAT; SO_4_^−^ was elevated in MAT compared to the other digesters. TSS and COD were substantially higher than at week six. Overall, at week six, wastewater characteristics were similar in all four digesters.

At week 42, biogas production was 189, 186, 233, and 235 L for CON, SND, MAT, and MST, with corresponding CH_4_ percentages of 65, 64, 66, and 65%. The pH of the digesters averaged 7.30; HCO_3_^−^ averaged 113 mM. NH_4_^+^ had increased to an average of 5.4 mM, while SO4^-^ averaged 120 µM. Overall, at weeks six and 23, wastewater quality was similar in the digesters, although at week 23, TSS was elevated in SND compared to the other digesters, likely indicating bubble “drag and lift” due to cavitational events in the sludge [33].

Although in general, wastewater and biogas characteristics were similar in all four digesters, substantially more biogas was produced by the treatments over a 66-week period: 17%, 28%, and 32% more by SND, MST, and MAT than by CON [20].

### 3.2. Description of Sequencing Outputs

Sequencing of 12 DNA samples extracted from the poultry litter digestate samples obtained from anaerobic digesters generated 243.7 million total reads with an average of 20.3 million reads per sample (range: 8.0 to 26.1 million reads). After trimming, 37.3% of the reads across all samples were removed. The remaining 152.9 million total trimmed reads with an average of 12.7 million reads per sample (range: 5.5–17.1 million) were classified as host (4.6% of reads), microorganisms (29.0% of reads), or unclassified (68.0% of reads). Table 2 presents an overview of changes in microbial populations in terms of taxonomic units due to time and treatment at the domain through species level.

### 3.3. Microbiome of Poultry Litter and Digestate Seed

The microbiome of PL was dominated by three species of Actinobacteria: *Brevibacterium intestinavium*, *Brachybacterium meravium*, and *Yaniella excrementigallinarum*, which comprised approximately 80% of reads (Appendix A). The next most abundant bacteria were the Firmicutes *Salinicoccus meravium* and *S. stercoripullorum* at 4–5% of OTUs each. No methanogens were found; and the only Archean detected was the halophile *Halococcus thailandensis* at an average relative abundance of 0.01%.

In week zero digestate, which consisted of a mixture of 600 g PL and 20 L seeds from from the commercial mesophilic anaerobic digester diluted to a total volume of 133 L, the only methanogens detected were *Methanoculleus bourgensis* and *Methanosarcina* sp002499445. Thus, the majority of future growth in Archaea, as shown below, came from the digester seed.

The digestate at week zero was dominated by Actinobacteria, Bacteroidetes, and Proteobacteria, comprising average abundances of 3.2%, 8.9%, and 84.2% of the reads, respectively. The most prominent species were the Proteobacteria *Acineobacter idrijaensis*, *A. lwoffi* D, and *Giesbergeria metamorpha* at 17%, 11.8%, and 9.1% of the reads.

### 3.4. Microbiome Dynamics over Time

The microbiome of the digesters was further analyzed and contrasted at weeks six, 23, and 42 because by week six, biogas production was starting to increase rapidly as was wastewater pH, and stable HCO_3_^−^ buffering was developing. At week 6, the microbiome of the four digesters was largely dominated by Actinobacteria (Actinomycetota) at an average of 39.7 ± 12.1% of the population, Bacteroidetes (Bacteroidota), with 6.9 ± 4.1%, Proteobacteria (Pseudomonadota), 23.9 ± 5.5%, and Firmicutes (Bacillota), which represented 20.5 ± 6.4% of sequences in the described population (Figure 1). At week 23, Actinobacteria had declined to 1.1 ± 0.6%, Bacteroidetes increased to 40.7 ± 6.6%, Proteobacteria represented 9.1 ± 2.8%, and Firmicutes comprised 11.9 ± 2.0% of the described population. At week 6, the most prominent Actinobacteria families were Micrococcaceae, Brevibacteriaceae, Dermabacteraceae, and Ruaniaceae, which comprised an average of 12%, 8.7%, 8.6%, and 6.9% of all OTU, respectively. Mycobacteriaceae, which occurred at approximately 3% of the reads in PL, comprised only 0.08% of all reads in week six digestate. The most abundant species of Actinobacteria in PL were *Yaniella excrementigallinarum* (18.8%), *Brachybacterium merdavium* (19.8%), and *B. merdavium* (19.7%), while the most abundant species in week 6 digestate were the Micrococcales Micrococcaceae bacterium UBA338 (11.7%) and *Ruania albidiflava* (6.6%); thus, despite Actinobacteria being prominent components of the microbiome in PL and week six digestate, the populations of Actinobacteria were distinctly different.

At week 42, Actinobacteria had increased to 12.3 ± 2%, Bacteroidetes were at 21.4 ± 6.4% of described species, Proteobacteria were 3.0 ± 0.6%, and Firmicutes were 27.8 ± 9.0% of the reads. Thus, on average, these four phyla composed 91% of the microbiome at week six, 63% at week 23, and 64.5% at week 42, indicating increasing microbial diversity as biogas production increased.

Nonetheless, in samples taken at week six, just prior to beginning the aeration and sound treatments, species diversity was high, with 213, 196, and 196 species found in CON, SND, and MAT digesters, respectively. Species diversity in MAT was lowest at week six with 58 species detected, likely due to low DNA yield, but was comparable to the other digesters at weeks 23 and 42 (Figure 2).

Actinobacteria exhibited greater diversity at week six than at weeks 23 and 42. At week six, 45 species belonging to nine orders were found in CON, 56 species belonging to nine orders were detected in SND, and 52 species belonging to nine orders were found in MAT. Only seven species of Actinobacteria belonging to two orders were found in MST at week six, but, as stated, species diversity in this digester increased with time and was comparable to the other digesters at weeks 23 and 42. Actinobacteria are phylogenetically and physiologically diverse and although some species are anaerobes, most are aerobes, and are important decomposers in soils, often forming filamentous structures resembling hyphae. Many are symbiotic with herbivores, aiding with the decomposition of plant polysaccharides [34,35]. Species of Actinobacteria comprised an average of 39.7%, 11.4%, and 9.4% of all out across the four treatments at week six, but only 1.1% and 12.3% at weeks 23 and 42, respectively. Despite the steep decline in relative abundance of Actinobacteria at week 23, by week 42 the relative abundance had increased to 12.3% and diversity was high with an average of 34.5 species in each of the four digesters. At week six, dominant bacteria included Micrococcales, especially an unknown Micrococcaceae bacterium UBA3381, *Ruania albidiflava*, and an unknown bacterium belonging to the genus *Brachybacterium*. The week 42 microbiome showed an increase in diversity in the order Streptosporangiales, with *Nocardiopsis xinjiangensis* and *N. synnemataformans* occurring in the greatest abundance.

At week 42 there was also a great increase in the abundance of in the family Mycobacteriaceae, order Mycobacteriales in the phylum Actinobacteria. The majority of these belonged to the genus Mycolicibacterium, some members of which are known to cause human disease but usually are saprophytic. They have been found in the cellulose-degrading consortium of an anerobic digester [36]. We thought it possible that the increase in the relative proportion of Actinobacteria at week 42 compared to week 23 was due to the increase in amount of PL being fed, since 1.0 kg wk^−1^ was being fed rather than the 600 g wk^−1^ at week 23, but while the most abundant Actinomycetes in PL were *Brachybacterium merdavium* (41.3%), *Brevibacterium intestinavium* (19.8%), and *Yaniella excrementigallinarum* (18.8%), the most abundant bacteria in week 42 digestate were three species of *Nocardiopsis*, in the order Streptosporangiales: *N. kunsanensis* (2.4%), *N. xinjiagensis* (1.4%), and *N. synnemataformans* (1.3%). None of the dominant PL species were detected in week 42 digestate. Thus, the increase in Actinobacteria at week 42 is indicative of progressive changes in the digester microbiome rather than being representative of the PL feed.

The Proteobacteria exhibit great metabolic diversity and include anaerobes, facultative anaerobes, phototrophs, chemolithotrophs, and heterotrophs. Some are thought to contribute to gut health by consuming O_2_, but are also implicated in disease due to an unbalanced gut microbiome [37]. In an anaerobic digester-treated waste active sludge, Guo et al. [38] found Proteobacteria comprised 41.5% of the population. In our digesters, Proteobacteria comprised 23.9%, 9.1%, and 7.5% of the microbiome at weeks 6, 23, and 42, respectively, highlighting the dynamic shifts in digester microflora that occurred. *Rhodopseudomonas faecalis* was the most dominant Proteobacterium, comprising 67.9, 49.6, and 37.4% of all Proteobacteria in CON, 44.3, 52.0, and 42% of MST, as well as 49.6, 53.9, and 53.9% of MAT, at weeks 6, 23, and 42, respectively. *R. facecallis* comprised 29.0 and 39.1% of all Proteobacteria in SND at weeks 23 and 42, while *R. palustris* comprised 75.4% of all Proteobacteria at week 6. Rhodopseudomonas are purple non-sulfur bacteria that are described as being capable of growing anaerobically in the light or aerobically in the dark using a variety of electron donors, and *R. faecillis* has been isolated from chicken feces digestate [39]. Rhodopsedomonas comprised 18.5% of all OTUs at week 6, but then underwent a steep decline, averaging 4.7% of OTU at week 23, and 1.2% at week 42. After Rhodopseudomonas spp., the next most abundant Proteobacterium present at week six was Bosea thiooxidans, with its sequences representing 1.0% of all prokaryotic OTUs.

Sulfate-reducing bacteria belonging to the order Desulfobacterales (Thermodesulfobacteriota, formerly Pseudomonadota) were prominent at week 6, representing approximately 4.3% of all sequences in the four digesters. Their abundance had declined to about 0.9% at week 42. While the relative abundance of Desulfobacterales decreased, their diversity varied with time. *Desulfovibrio spp*. represented 91–100% of Desulfobacterales at week 6, but only 62% at week 23, and 17% at week 42. This decline in relative abundance was due to an increase in the relative abundance of *Desulfobulbus elongatus* and *D. propionicus*. While traditionally classed in the Proteobacteria, they are now placed in the phylum Thermodesulfobacteria [40]. Other SO_4_^2−^ reducers such as Clostridiales belonging to the genera *Desulfallas*, *Desulfibacter*, and *Desulfosporosinus* occurred at low abundances of less than 0.05%. Perhaps the decline in relative abundance of Desulfobacterales was due to the relatively low concentration of SO_4_^2−^ in the digestate and/or competition for H_2_ with methanogens as the abundance of the latter increased. Sulfate concentrations averaged 31 µM in the digesters at week 6, and only 38.5 µM at week 23. At week 42, SO_4_^2−^ concentrations were lower still, averaging 130 µM. This would have served to reduce the SO_4_^2−^-reducing bacteria’s competitive advantage for H_2_ vis-à-vis methanogens [41]. Still, it is interesting to note that SO_4_^2−^ concentrations were higher in MAT and MST (Table 1), perhaps indicating some additional sulfur oxidation in the aerated digesters.

The phylum Bacteroidota averaged 6.3%, 40.6%, and 21.4% of all OTU at weeks 6, 23, and 42, respectively, with the order Bacteroidales accounting for 97% or higher of all Bacteroidetes, regardless of sampling time or digester. However, species diversity of Bacteroidetes increased over time, with an average of 16.75 species found in each digester at week 6, with an average of 55 and 39.75 species identified at weeks 23 and 42, respectively. Most of these species belonged to the order Bacteroidales, which at week six accounted for over 90% of Bacteroidetes species, and about 84% of all Bacteroidetes at both weeks 23 and 42, averaged for all four treatments. At week six, 36.5% of all Bacteroidetes belonged to the genus *Bacteroides*, which declined to 5.4% at week 23 and 3.4% by week 42. The decline in the relative abundance of *Bacteroides* after week 6 was largely due to large increases in the relative abundance of an unknown bacterium, Bacteroidales bacterium UBA5326, which comprised an average of 19.5% of enumerated prokaryotes in all four digesters at week 23 and 7.7% at week 42.

Bacteroidota are well known polysaccharide degraders in the environment and the gut microbiome. It has been suggested, based on phenotype-based clustering in the Bacteroidetes, that the ability to degrade specific polysaccharides varies greatly even among closely related species or even strains. Therefore, efficient catabolism of complex mixtures of polymers including carbohydrates occurs most efficiently in a diverse community of bacteria that vary widely in their ability to degrade specific classes of polysaccharides [42,43]. Therefore, the increase in species richness of Bacteroidetes in the digesters would likely improve their ability to degrade cellulose and other complex β-glycans associated with PL.

Species diversity for the Firmicutes was high with an average of 91.9 species identified in samples taken on the three sampling dates. Unlike the Bacteroidetes, however, species diversity did not markedly increase until week 42, with an average of 77, 77.5, and 123.25 species identified per digester at weeks 6, 23, and 42, respectively. The greatest increase in species richness occurred in the family Bacilliaceae, which increased from an average of 4.25 species per digester at week 6 to an average of 27 species per digester at week 42. The Bacillaceae are facultative anaerobes. At least one species identified in the present study, *Bacillus cereus*, seems to preferentially degrade proteins rather than carbohydrates as a nutrient source [44].

Other prominent genera of Firmicutes included *Clostridium, Acetobacterium, Eubacterium*, and unknown species belonging to the family Ruminococcaceae. Clostridia are obligate anaerobes and important plant polysaccharide degraders and fermenters [45], while *Acetobacterium* are acetogens that generate a sodium rather than a proton gradient to generate ATP and reduce CO_2_ to acetyl-CoA [46].

Twelve species belonging to the order Spirochaetes were found in the digesters with three identified species: *Sphaerochaeta globosa, S. halotolerans*, and *S. pleomorpha*. None of the Spirochaetes were detected in any of the digesters at week 6, whereas they comprised 4.6%, 6.3%, 3.4%, and 2.4% of all enumerated species at week 23 for CON, SND, MAT, and MST, respectively, and 2.7%, 1.6%, 2.1%, 0.8% of enumerated species at week 42. Lee at al. [47] found that the addition of acetate to digesters stimulated Spirochaetes, suggesting that they were involved in syntrophic acetate oxidation, i.e., oxidizing acetate to H_2_ and CO_2_ or formate so that the products can be used by a syntrophic partner such as a methanogen or sulfate reducer. Tokuda et al. [48] found Spirochaetes to be primary degraders of xylans, a type of hemicellulose, in the hindgut of *Nasutitermes takasagoensis*, a wood-feeding termite, while Ransom-Jones et al. [49] found that Spirochetes, along with Firmicutes, Bacteroidetes, Spirochaetes, and Fibrobacteres belonged to a community of cellulose degraders in landfills. These phyla were found in great diversity in the digesters except for Fibrobacteres, where only one unidentified species was found, Fibrobacteria bacterium AD312. The relative abundance of this bacterium increased from 0.01% at week six to 0.08% of all OTU at week 42. Species in the genus *Fibrobacter* were transferred from the genus *Bacteroides* in 1988 [50]. They are known cellulose degraders in the rumen [51]. The tendency for an increase in diversity of OTU with time is almost certainly driven not only by slow growth rates of many species that were undetectable at week six, but also by a greater diversity and quantities of substrates available for utilization by fastidious species.

At week six, methanogen diversity was low, with an unknown species of *Methanobrevibacter* and *Methanosarcina mazei* and *M. spelaei* comprising over 95% of Archaea sequences. By week 23, the genera *Methanosarcina, Metanothrix*, and *Methanospirillum* comprised about 98% of Archaea sequences, but at week 42, these four genera plus *Methanoculleus* and an unknown genus in the family Methanoregulaceae comprised approximately 95% of the Archaea. Twenty-nine distinct species of methanogens were identified, the major genera of which are presented in Figure 3.

Figure 4 represents prokaryotic phyla and species richness of the phyla at weeks 23 and 42, at which point sound- and air-treatment of the digesters had been performed for 17 and 37 weeks, respectively. Compared to week six, species richness had increased by 4%, 5%, and 31% over their week six abundances for CON, MAT, and SND, respectively, by week 23, and by a further 6%, 27%, and 16% by week 42. Although species richness increased to much greater degrees in SND and MAT than in CON, there was no remarkable taxonomic shift in the composition of the microbiomes that could be attributed to air or sound treatment.

More so than causing a dramatic shift in the relative proportions of microbial taxa in the digestate, treatment with air or sound seemed more so to increase the overall diversity of the microbial biome, with 207, 257, and 233 species identified in CON, SND, and MAT, respectively, at week 23, while at week 42, 237 were identified in CON, and 296 identified in both SND and MAT. Diversity, gauged by Shannon indices, was 3.69, 3.83, and 3.57 for CON, SND, and MAT at week 23, and 4.05, 4.37, and 4.34 at week 42. Possessing the greatest number of species, this effect was most pronounced in the phylum Firmicutes at week 42 with 98 species identified in CON at week 42, and 129 and 148 identified during the same week in SND and MAT. The overall abundance of Firmicutes OTU at week 42 was 13.5%, 26.8%, and 28.1% for CON, SND, and MAT, respectively.

At a lower level of classification, this might be most clearly demonstrated with *Clostridium*. At week 23, one species of *Clostridium* was found in CON, comprising 0.33% of all OTUs, while at week 42, six were found comprising 0.45% of all OTUs. In SND, two species comprising 0.11% of OTUs were found at week 23, and 10 *Clostridium* species comprising 0.63% of OTU were found at week 42. In MAT, four species comprising 0.37% of OTUs were detected at week 6, and 10 species containing 2.63% of all prokaryotic OTUs were found at week 42. Diversification of species in the digesters could result from the growth of previously undetected species with inherently slow growth rates, the sudden growth of species due to temporal niches appearing in the digester due to changes in biotic or abiotic conditions, or the growth of fastidious organisms exploiting required nutrients released by the activity of other species.

Thus, both sound treatment and microaeration served to enhance microbial diversity and ultimately enhance biogas production from poultry litter. As noted with the human gut microbiome, a more diverse community is not only functionally diverse but also functionally redundant, perhaps more likely to cope with structurally complex and temporally variable feedstocks as well as environmental perturbations such as shifts in redox potential or increases in toxins such as ammonia [52,53,54,55].

It seems reasonable that increased microbial diversity would also be reflected by enhanced microbial biomass, although no quantitative measure of this was performed. Greater nutrient availability has been shown to drive effect increases in both microbial biomass and alpha diversity in the guts of arctic ground squirrels since species diversity and bacterial density were lower in hibernating animals than in active squirrels [56]. While more microbial biomass could largely account for the enhanced biogas production by the sound and aeration treatments, improving microbial diversity should act to increase digester stability since a diverse microbiome would be more capable of adjusting to variations in temperature, pH, ammonium, redox potential, and other factors given potential sensitivity of individual species to environmental variation. According to Kirchmann [57], “A diverse, functionally redundant microbial community may respond more quickly than a less diverse community to perturbations”.

### 3.5. Interpretation of Results in Reference to Previous Research

The comparison of dosing of microaeration between experiments is difficult due to variation in metrics used to report air delivery, i.e., delivery of aeration based on grams of volatile solids or grams of soluble chemical oxygen demand and wide variation in digester design, e.g., batch reactors, continuous flow reactors, upflow anaerobic sludge blanket reactors, etc. Also, there is variation in air delivery techniques such as injection in the headspace and bubbling into the digestate or even electrolytic aeration where O_2_ and H_2_ are simultaneously supplied to digesters [58]. Readers are referred to reviews for in depth discussions on the current state of microaeration technology [59,60,61].

The most easily translatable metric used is volume of air used per L of digestate, although this metric does not consider digestate strength nor accumulated volatile solids. In our experiment, aeration was delivered every 6 h in 200 mL doses into a subsurface manifold designed to aid in retention of the aeration within the sludge layer. This represented a nominal O_2_ dose of 42 mg O_2_ four times daily affording approximately 6 mL air L^−1^ digestate d^−1^ or 1.26 mg O_2_ L^−1^ d^−1^ digestate. This dosing was used because, in a previous study, delivery at 1.5 mL L^−1^ digestate d^−1^ stimulated biogas production little compared to an unaerated digester and dosing at 15 mL air L^−1^ digestate d^−1^ reduced biogas production by 19% [21].

Lim and Wang pretreated a brown water and food waste mixture with 37.5 mL O_2_ L^−1^ for four days to achieve a 10–20% increase in CH_4_ yield [62]. They postulated that this increase could be due to an increase in hydrolytic and acidogenic bacteria. Microaeration of blackwater at high doses has been shown to increase the abundances of facultative anaerobic bacteria (Ignavibacteriales and Cloacamonales) and aerobic bacteria (Rhodocyclales) compared to unaerated blackwater [63]. Still, our study used relatively less air, where for instance, in the latter study, air was injected into the blackwater four times over the course of 103 days at doses of 0, 5, 10, 50, or 150 mg O_2_ mg^−1^ COD. The microbiome of low O_2_ doses (0–10 mg) was characterized by fermenting and syntrophic bacteria as well as methanogens, whereas high doses (50–150 mg O_2_) had more facultatively anaerobic bacteria and aerobic bacteria. At 800 mL air d^−1^, our study dosed the digesters at roughly 9.5 µg O_2_ mg^−1^ COD wk^−1^ at week 6 and 1.8 µg O_2_ mg^−1^ COD wk^−1^ at week 42, so it is not surprising that we did not note similar increases in facultative anaerobes. Still, we did note the COD of both digesters treated by microaeration was lower than that of CON and SND (Table 1).

It is probable that the design of the microaeration apparatus accounted for the perceived lack of systematic changes in microbiome composition, such as those noted in previous research [62,63]. Rather, the composition of the microbiome was shifted in more subtle ways. There was a noticeable increase in species diversity. This was probably due to the aeration manifold being designed to keep aeration within the sludge layer. If the growth of aerobes or facultative anaerobes was fostered by the microaeration, they would be preferentially located within the sludge layer, likely occurring within the manifold as sessile cells and less likely to be noted in suspended solids or as planktonic species. Similarly, no systemic changes in the digestate microbiome could plainly be attributed to sound treatment, but the Shannon index at week 42 was considerably higher at 4.37 for ST than that of CON at 4.05. Species richness at week 42 was 237 for CON, and 296 for SND and 298 for MAT. Both microaeration and sound treatment could serve to enhance nutrient availability and account for observed changes in species diversity and richness.

In an experiment prior to the present study [21], aeration rates ranging from 200 mL d^−1^ to 2000 mL d^−1^ were tested to determine the rate of 800 mL d^−1^ used in the present study. The digesters had disks composed of *Liriodendron tulipfera* wood placed in the tanks. At the end of the experiment, a white biofilm with a myceliar appearance was found growing on the surface of the digestate and wood disks in digesters receiving 800 mL d^−1^ and 2000 mL d^−1^, but not in the anaerobic digester or the digester receiving 200 mL d^−1^. Subsequent analyses of wipes taken of the surface of the wood disks from the anaerobic digester and the digester receiving 800 mL air d^−1^ were sequenced. The results of these analyses are presented in the supplemental material. The wood disks showed pronounced differences in the microbiomes found on their surface. On the anaerobically cultured disk, the most prominent Archaea were *Methanosarcina* spp., occurring at 6.8% of all prokaryotic OTU, while on the surface of the wood disk in the aerated digester, *Methanosarcina* spp. occurred at an abundance of 54.3% of all OTUs. *Methanosarcina* are the only methanogens that are known to be capable of producing CH_4_ by all three metabolic pathways, that is, hydrogenotrophic, acetoclastic, and methylotrophic [52,54,64].

Proteobacteria comprised 79.4% of all prokaryotes on wood disks in the anaerobic digester, but only 6.3% in the aerated digester. Actinobacteria comprised 3.9% of all OTU on disks in the anaerobic digester and 16.3% in the aerated digester. Pronounced differences were also noted in the relative abundances of Firmicutes, with this phylum comprising 1.8% of OTU on the wood disks in the anaerobic digester and 9.2% in the aerated digester. Bacteroidota occurred at 2.8% and 4.7% of OTU in the anaerobic and aerated digesters, respectively.

Firmicutes and Bacteroidota are considered the most important hydrolytic bacteria driving anaerobic digestion [65,66], though Actinobacteria are also well-known decomposers of plant polysaccharides [34,35,67]. Although it is likely that the microbiome found on the wood disks in the previous experiment differed substantially from that of the sludge, the elevated abundances of Firmicutes, Bacteroidota, and Actinobacteriota on aerated wood disks shows that the microaeration treatment was capable of greatly altering the microbiome of the wood disks and likely accelerating polysaccharide hydrolysis. This is particularly noteworthy considering that the wood disks probably received little to no aeration due to the manifold being designed to retain air within the sludge layer.

It is also worth noting that in the present study, the clearest disparity in digestate microbial abundances was seen in the Firmicutes, where they occurred at an abundance of 13.5% of OTU in CON, and 26.8%, 28.1%, and 37.6% of OTU in SND, MAT, and MST, respectively. Differences in the relative abundances of Bacteroidota and Actinobacteria were less pronounced. Further research studying the effect of sound and microaeration on the microbiome of PL sludge is desirable.

In retrospect, if combined aeration with sound did cause the formation of reactive species such as ^*^OH, ^*^H, and ^*^O_2_^_^ [22,23], timing of sound treatment and air delivery should have been staggered to prevent or lessen the adverse effects of the combined treatment. Since air was delivered in 200-mL aliquots every six hours, waiting until a considerable portion of the oxygen had been consumed before treatment with sound might have reduced the chances of forming free radicals.

The literature on the use of low frequency sound as opposed to ultrasound and its effect on microbial growth is sparse. In one experiment, biomass and growth rate of *Escherichia coli* k-12 exposed to an 80 dB, 8000 Hz sound wave were 1.7- and 2.5 times higher than that of nonexposed *E. coli*, respectively [68,69]. Harris et al. showed that a 90 dB, 100 Hz, or 10 kHz sound wave increased the production of volatile metabolites and cell growth of the Ascomycete *Saccharomyces cerevisiae* [70]. In addition to the present research, we have demonstrated enhanced biogas production by an 800-L digestate of corn meal and a digester with a digestate volume of 9.1 m^3^ containing mixed waste digestate using various sine wavelengths and music all below 20 kHz in frequency, but generally below 5 kHz in frequency [20]. The sound-induced physical mechanisms that are plausibly responsible for enhanced gas production and increased species diversity are numerous and include inertial cavitation that could help disrupt sludge, acoustic streaming, which is a bulk flow phenomenon that could aid in nutrient mixing, sonoporation, which enhances cell membrane porosity, and disruption of the bacterial cell boundary layer, facilitating exchange between bacteria and the environment [18,19,20,71,72,73].

Mechanical vibration of the sludge could also serve to enhance biogas and cell proliferation. Mechanical vibration, in addition to aiding nutrient exchange, could help solubilize recalcitrant β-linked polysaccharides. Cellulose, for instance, is more resistant to enzymatic and acid hydrolysis than are starches and many other plant polysaccharides. This is due to its crystalline nature, which arises from hydrogen bonding between individual polymer chains forming larger fibrils that ultimately assemble into water insoluble fibers. While starches are also somewhat crystalline in character, water more easily penetrates between the individual polymer chains than is the case for cellulose, rendering starches more susceptible to chemical and biochemical hydrolysis [74]. Cellulose degradation, therefore, occurs extracellularly, on the bacterial cell surface, and makes carbon available for other members of the microbial community. Nevertheless, it proceeds much more slowly than in aerobic environments [75,76]. Ultrasonification, however, has been shown to promote the breakage of larger cellulose fibers into smaller fibrils, making them less resistant to hydrolysis [77]. It is likely that sound treatment at lower frequencies also has this effect.

At week 42, Actinobacteria comprised 14.8% of OTU in digestate of CON and 14.9% of OTU in SND. While during the same week, Bacteroidota comprised 30.8% of OTU in CON and 20.3%, 12.7%, and 21.9% of OTU in SND, MST, and MAT, respectively. Relative abundances of Proteobacteria ranged from 2.8% for CON to 3.8% for SND. Firmicutes, however, occurred 17.4%, 23.0%, 41.6%, and 29.0% of OTU in CON, SND, MST, and MAT, respectively. The relatively high abundance of Firmicutes in MST and MAT suggests possible resistance to damage caused by free radicals formed in cavitation events. Therefore, although we did not note marked changes in the composition of the microbiome due to the treatments, the changes were most distinct in shifts of the relative percentages of Bacteroidota and Firmicutes.

More obviously, microbial diversity was enhanced by both sound and microaeration treatments. It seems reasonable that increased microbial diversity would also be reflective of increased microbial biomass, as previously noted in Arctic ground squirrels [56]. While greater microbial growth would account for the enhanced biogas production by the sound and aeration treatments, greater diversity should act to increase digester stability since a diverse microbiome would be more capable of adjusting to variations in temperature, pH, ammonium, and other factors given differences in the response of individual species to these variations.

## 4. Conclusions

The diversity of the prokaryotic microbiome was enhanced in the PL digestate due both to time and treatment with microaeration or sound treatment. At week 23, average species richness increased 38% over that of week 6, and by a further 15% at week 42. At week 23, species richness in SND, MAT, and MST was 24%, 13%, and 9% higher than that of CON, while at week 42, species richness in SND and MAT was 25% higher than that of CON. Polysaccharide-degrading bacteria, especially within the Firmicutes and Actinobacteriota, constituted a significant portion of this increase in the number of species. Species richness in MST at week 42 was 4% lower than that of CON, possibly due to interference of sound treatment with microaeration or due to oxidative toxicity caused by combining aeration with sound treatment.

Enhancing microbial diversity in anaerobic digesters is highly desirable to boost biogas production, increase digester stability, and enhance sludge degradation. Achieving these goals will foster the adoption of anaerobic digestion as a means of waste valorization.

## Figures and Tables

**Figure 1 microorganisms-11-02349-f001:**
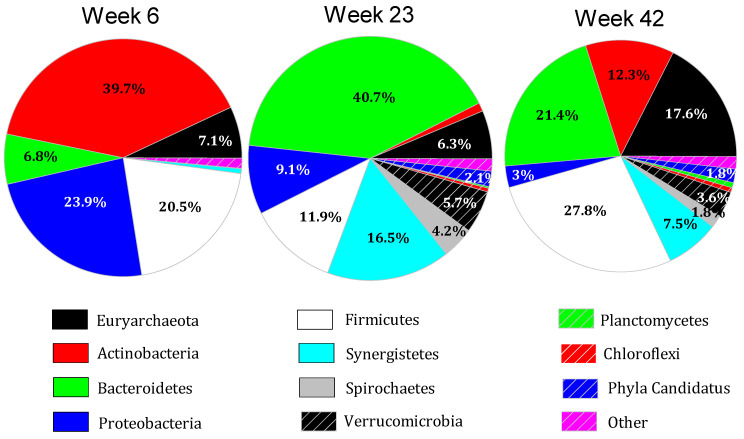
Major phyla present in the digesters at weeks six, 23, and 42. Proportions reflect the averages of all four digesters: control, sound-treated, air-treated, and air- and sound-treated. Phyla candidatus are proposed phyla Atribacteria, Cloacimonetes, Hydrogenedentes, Omnitrophica, Riflebacteria, and Sumerlaeota.

**Figure 2 microorganisms-11-02349-f002:**
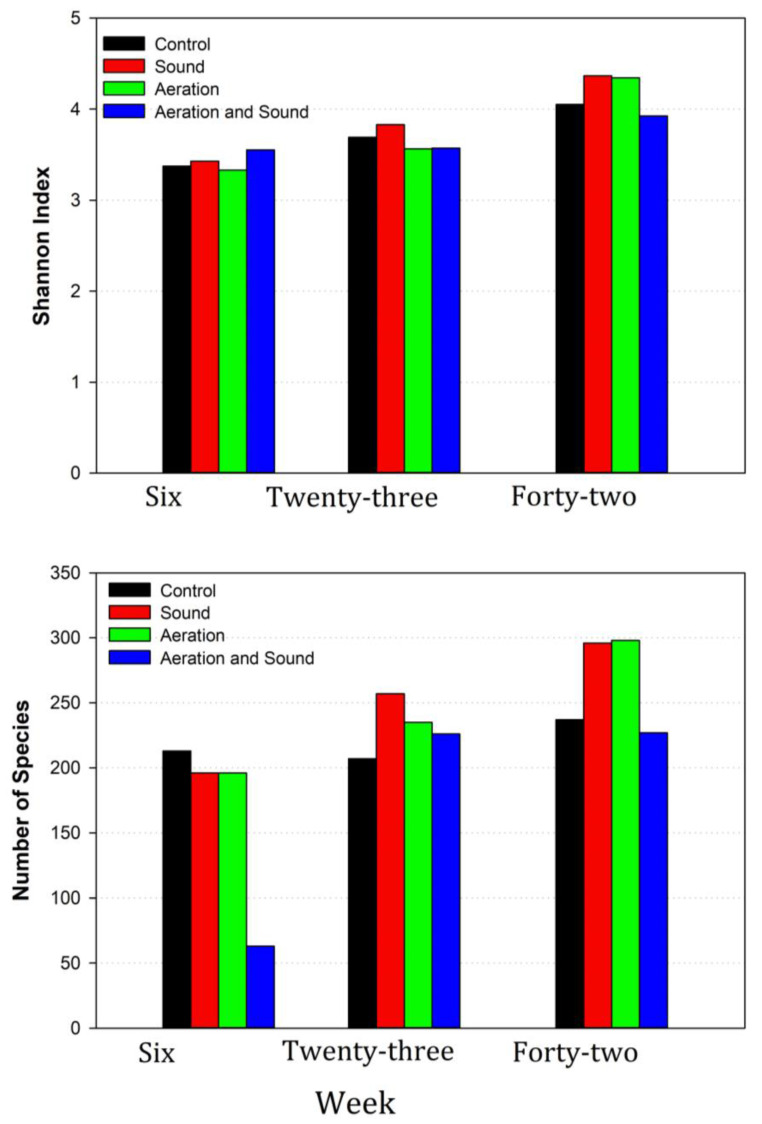
Indices of species diversity and richness at weeks six, 23, and 42. Data represent the average Shannon index and the number of species detected for the four digesters.

**Figure 3 microorganisms-11-02349-f003:**
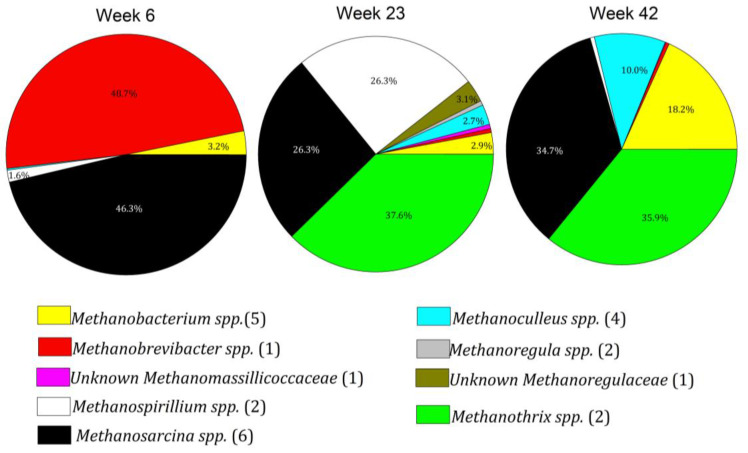
Relative abundance of methanogenic genera as a percentage of Euryarchaeal sequences. Number of species in each genus in parenthesis.

**Figure 4 microorganisms-11-02349-f004:**
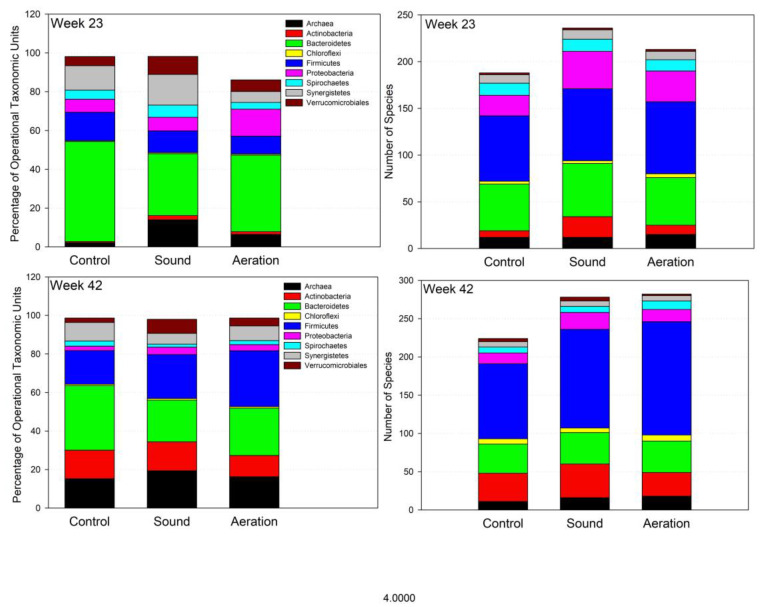
Major prokaryotic phyla as a percentage of operational taxonomic units and phylum species richness at weeks 23 and 42 for the control, sound-treated, and aerated digesters.

**Table 1 microorganisms-11-02349-t001:** Biogas and digestate characteristics during weeks samples were taken for microbiome analyses.

	Treatment			
	Control	Sound	Aeration	Sound with Aeration
	Week 6			
	Biogas Characteristics			
CO_2_ (µg L^−1^)	638,000	629,000	660,000	638,000
CH_4_ (µg L^−1^)	274,000	435,000	451,000	407,000
Biogas Volume (L wk^−1^)	50.0	44.3	49.2	53.1
Millimoles CH_4_ wk^−1^	855	1200	1380	1350
	Digestate Characteristics			
pH	6.45	6.48	6.54	6.50
HCO_3_^−^ (mM)	15.3	15.4	19.5	17.2
NH_4_^+^ (mM)	1.79	1.74	1.51	1.74
SO_4_^−^ (µM)	29.5	28.0	31.6	35.1
Chemical oxygen demand (mg L^−1^)	1140	1290	1200	1160
Total suspended solids (mg L^−1^)	59	50	61	50
	Week 23			
	Biogas Characteristics			
CO_2_ (µg L^−1^)	594,000	629,000	660,000	638,000
CH_4_ (µg L^−1^)	409,000	415,000	435,000	441,000
Biogas Volume (L wk^−1^)	144	166	188	172
Millimoles CH_4_ wk^−1^	3600	4300	5090	4730
	Digestate Characteristics			
pH	7.41	7.41	7.39	7.38
HCO_3_^−^ (mM)	87.3	84.8	84.3	83.6
NH_4_^+^ (mM)	2.26	2.17	3.04	2.13
SO_4_^−^ (µM)	31.4	26.3	68.3	27.9
Chemical oxygen demand (mg L^−1^)	3420	3820	3130	3170
Total suspended solids (mg L^−1^)	160	568	211	183
	Week 42			
	Biogas Characteristics			
CO_2_ (µg L^−1^)	691,000	603,000	651,000	533,000
CH_4_ (µg L^−1^)	472,000	460,000	467,000	427,000
Biogas Volume (L wk^−1^)	248	286	322	309
Millimoles CH_4_ wk^−1^	6950	7120	5280	5660
	Digestate Characteristics			
pH	7.32	7.32	7.26	7.31
HCO_3_^−^ (mM)	108	118	146	130
NH_4_^+^ (mM)	5.52	2.17	3.05	2.13
SO_4_^−^ (µM)	95.9	156	66.3	201
Chemical oxygen demand (mg L^−1^)	6950	7120	5280	5660
Total suspended solids (mg L^−1^)	877	1530	510	737

**Table 2 microorganisms-11-02349-t002:** Number of taxonomic units found in digesters at weeks six, 23, and 42.

	Control			Sound			Microaeration			Microaeration and Sound		
Week	6	23	42	6	23	42	6	23	42	6	23	42
	Archaea											
Orders	3	4	3	3	4	4	3	3	5	3	4	3
Families	3	7	5	3	6	6	4	7	8	3	6	5
Genera	3	8	6	4	7	7	4	6	6	5	8	9
Species	4	12	11	7	12	16	10	15	18	5	10	10
	Bacteria											
Orders	37	42	44	37	50	43	32	43	39	18	47	34
Families	77	70	79	75	94	97	67	77	81	34	80	74
Genera	131	107	123	115	136	157	114	120	147	38	117	121
Species	209	195	226	189	245	280	186	220	280	58	217	217

## Data Availability

The following are available online at http://www.mdpi.com/: 1. Sample key.xlsx: Excel spreadsheet containing group comparisons for all microbiome analyses. Sequence data were deposited into National Center for Biotechnology Information (NCBI) under Sequence Read Archive (SRA) and can be accessed by the BioProject ID: PRJNA1013451 with the following BioSample Accession numbers: SAMN37298551, SAMN37298552, SAMN37298553, SAMN37298554, SAMN37298555, SAMN37298556, SAMN37298557, SAMN37298558, SAMN37298559, SAMN37298560, SAMN37298561, SAMN37298562, SAMN37298563, SAMN37298564, SAMN37298565, SAMN37298566, SAMN37298567, SAMN37298568, SAMN37298569, SAMN37298570, SAMN37298571.

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
