# Peer review of "Microbiome Diversity of Anaerobic Digesters Is Enhanced by Microaeration and Low Frequency Sound"

_microorganisms, 2023, doi:10.3390/microorganisms11092349_

Round 1
Reviewer 1 Report
General comments
-This manuscript, by authors, studied “
“Microbiome Diversity of Anaerobic Digesters is Enhanced by Microaeration and Low Frequency Sound”.
Overall, the topic is of interest to microorganisms, and readers. However, the following are the specific comments on the article concerns, before publication as minor revision.
-Abstract
-“At week 42, Firmicutes, Bacteroidetes, Euryarchaeota, and Actinobacteria were dominant.” How? Can add more results.
-“improve nutrient availability.” Not found such related content in the manuscript.
-Introduction
-Better to avoid short paragraphs. Can combine related ones.
-Specific objectives?
-Materials and Methods
-“The digesters have been previously described [20].” Can add little more details than a reference.
-In sections 2.2 and 2.3, please add references for methodology.
- Please add experimental setup (field and lab experiments), pictures, and more figures to make the manuscript more attractive.
-Results and Discussion
- “In the previous study [20].” ?
Without details about that previous study is not an appropriate way to start in the results section. Please improve this sentence structure.
- Please improve Table 1 structure.
- Table 2 also needs to be improved with specific names of species (Most significant results are enough. Any unit for sound detection? How do you measure the values of sound? And sound with aeration?
-Orders, Families, Genera, Species…. no details in the introduction section?
Be specific with major findings.
-Enlarge your figures for a better view.
Author Response
-“At week 42, Firmicutes, Bacteroidetes, Euryarchaeota, and Actinobacteria were dominant.” How? Can add more results.
We added the actual proportions of the phyla to this sentence. Page 1, lines 19-20.
“improve nutrient availability.” Not found such related content in the manuscript.
This sentence is speculative, we admit. We modified it to read: “Given that both air and sound treatment increased biogas production, it is likely that they improved poultry litter breakdown to promote microbial growth.” Page 1, lines 23-25.
-Introduction
-Better to avoid short paragraphs. Can combine related ones.
We combined paragraphs one and two of the Introduction as well as paragraphs five and six. We ask the Editor to decide this matter.
Specific objectives?
We modified the last sentence of the Introduction to read: “In this way, we hoped to gain insight into how engineering and operational modifications to digester operation affect the microbiome of digesters can lead to the development of more efficient and reliable designs that enhance biogas production. “ Lines 116-117
Materials and Methods
-“The digesters have been previously described [20].” Can add little more details than a reference.
I have added more details to paragraphs one and two to the Materials and Methods section.
-In sections 2.2 and 2.3, please add references for methodology.
We added a reference for the wastewater analyses (American Association for Public Health) and the references for microbiome analysis are included. (Lines 164-165)
- Please add experimental setup (field and lab experiments), pictures, and more figures to make the manuscript more attractive.
We have an illustration of the digester in reference 20 which we added to the manuscript. Due to copyright issues we believe we couldn’t add this illustration and couldn’t improve upon it in this manuscript.
Results and Discussion
- “In the previous study [20].” ?
Without details about that previous study is not an appropriate way to start in the results section. Please improve this sentence structure.
We moved this sentence to the end of the paragraph. (Lines 210-214)
- Please improve Table 1 structure.
Table 1 is large, but we feel this level of detail is desirable to give the reader knowledge of the environmental conditions during the times the samples were taken for analysis. We did change the units for CO2 and CH4 to micrograms per liter which is more common in the literature.
- Table 2 also needs to be improved with specific names of species (Most significant results are enough. Any unit for sound detection? How do you measure the values of sound? And sound with aeration?
In this experiment we did not measure the sound as we did in some other experiments (i.e. reference #19). We feel that Table 2 would become rather messy if included names of species found at specific weeks especially since few species were especially dominant during specific weeks.
-Orders, Families, Genera, Species…. no details in the introduction section? We briefly referenced the findings of some previous research but go into more detail in the results and discussion.
Be specific with major findings.
We tried to do this and tried to emphasize that the major finding of the research was a general increase in species diversity and richness.
-Enlarge your figures for a better view.
The figures were rather small, and we apologize for that. We were just trying to avoid large blank spaces on the pages as we are not professional document preparers. The original graphic files are high quality and were depending on the staff at Microorganisms to prepare the final version of the manuscript.
Reviewer 2 Report
There are some questions and remarks:
1. L70. CH4 index.
2. L117. Should be organic load, but not just mass per week to unknown volume of digester. Scientific indicators are usually specific but not general numbers.
3. L131. It is not clear what was oxygen saturation coefficient, was the complete aeration assured in whole volume, was mixing applied for substrate? This part must be cleared and described in more details.
4. It is not clear if during whole experiment the same sample of PL was used for all 42 weeks and all feeding quantities? Describe it somewhere around L150.
5. Table 1. Convert biogas volume, methane generation to internationally acceptable %, L/kg VSS and similar. Now we have biogas yield during week from some 133 L of volume. So it must be indicated in specific numbers in order to be comparable to other research results. As well, COD, TSS has no meas. units.
English is good.
Author Response
. L70. CH4 index. Was this referring to changing CH4 to CH4? We corrected this. (Line 70)
2. L117. Should be organic load, but not just mass per week to unknown volume of digester. Scientific indicators are usually specific but not general numbers.
In the Experimental section we fully describe the digester volume and organic loading rate (see L152-L157). In the introduction we were just trying to qualitatively state that the loading rate was changing during the experiment.
3. L131. It is not clear what was oxygen saturation coefficient, was the complete aeration assured in whole volume, was mixing applied for substrate? This part must be cleared and described in more details.
We are not sure what the reviewer’s question is here but have tried to rewrite the description of how the aeration was supplied to make it clearer. (Lines 131-135)